# Cell-Free DNA Genomic Profiling and Its Clinical Implementation in Advanced Prostate Cancer

**DOI:** 10.3390/cancers16010045

**Published:** 2023-12-21

**Authors:** Ivana Bratic Hench, Luca Roma, Floriana Conticelli, Lenard Bubendorf, Byron Calgua, Clémentine Le Magnen, Salvatore Piscuoglio, Mark A. Rubin, Alin Chirindel, Guillaume P. Nicolas, Tatjana Vlajnic, Tobias Zellweger, Arnoud J. Templeton, Frank Stenner, Christian Ruiz, Cyrill Rentsch, Lukas Bubendorf

**Affiliations:** 1Institute of Medical Genetics and Pathology, University Hospital Basel, University of Basel, 4031 Basel, Switzerland; 2Department of Public Health, University of Naples Federico II, 80131 Naples, Italy; 3Department of Urology, University Hospital Basel, University of Basel, 4031 Basel, Switzerland; 4Department of Biomedicine, University Hospital Basel, University of Basel, 4031 Basel, Switzerland; 5Precision Oncology Laboratory, Department for Biomedical Research, Bern Center for Precision Medicine, 3008 Bern, Switzerland; 6Bern Center for Precision Medicine, Inselspital, Bern University Hospital, University of Bern, 3008 Bern, Switzerland; 7Division of Nuclear Medicine, Department of Theragnostics, University Hospital Basel, 4031 Basel, Switzerland; 8St. Claraspital, 4058 Basel, Switzerland; 9St. Clara Research, Basel and Faculty of Medicine, University Basel, 4058 Basel, Switzerland; 10Division of Oncology, University Hospital Basel, 4031 Basel, Switzerland

**Keywords:** prostate cancer, liquid biopsy, precision medicine, circulating cell-free DNA, next-generation sequencing

## Abstract

**Simple Summary:**

Prostate cancer (PCa) accounts for the second-highest mortality rate in cancer-associated deaths. Liquid biopsy has great potential to tailor treatment choices in advanced PCa. In this study, we evaluated the utility and diagnostic potential of a custom-designed NGS cfDNA panel based on the AmpliSeq HD Technology in advanced PCa. cfDNA somatic mutations were detected in the majority (71%) of examined advanced PCa patients. The most frequently mutated genes in the PCa cohort of 68 patients (40 metastatic castration-resistant and 28 metastatic hormone-naive PCa) were: *TP53*, *FOXA1*, *SPOP*, *PTEN*, *AR*, *CTNNB1*, *RB1*, and *PIK3CA*. *AR* amplifications were detected in 31% of mCRPC patients. This approach appears to be a straightforward and cost-effective method for detecting clinically relevant somatic mutations in cfDNA to aid in clinical decision-making.

**Abstract:**

Most men with prostate cancer (PCa), despite potentially curable localized disease at initial diagnosis, progress to metastatic disease. Despite numerous treatment options, choosing the optimal treatment for individual patients remains challenging. Biomarkers guiding treatment sequences in an advanced setting are lacking. To estimate the diagnostic potential of liquid biopsies in guiding personalized treatment of PCa, we evaluated the utility of a custom-targeted next-generation sequencing (NGS) panel based on the AmpliSeq HD Technology. Ultra-deep sequencing on plasma circulating free DNA (cfDNA) samples of 40 metastatic castration-resistant PCa (mCRPC) and 28 metastatic hormone-naive PCa (mCSPC) was performed. CfDNA somatic mutations were detected in 48/68 (71%) patients. Of those 68 patients, 42 had matched tumor and cfDNA samples. In 21/42 (50%) patients, mutations from the primary tumor tissue were detected in the plasma cfDNA. In 7/42 (17%) patients, mutations found in the primary tumor were not detected in the cfDNA. Mutations from primary tumors were detected in all tested mCRPC patients (17/17), but only in 4/11 with mCSPC. AR amplifications were detected in 12/39 (31%) mCRPC patients. These results indicate that our targeted NGS approach has high sensitivity and specificity for detecting clinically relevant mutations in PCa.

## 1. Introduction

Prostate cancer (PCa) accounts for the second highest mortality rate in cancer-associated deaths in men after lung cancer [1]. While most patients present at a localized stage of the disease, some are diagnosed with locally advanced or metastatic disease (mPCa). There are several therapeutic options for the treatment of mPCa available, and the identification of the predictive biomarkers that would guide clinical decision-making in selecting effective treatment strategies is of high clinical relevance. There are two challenges in the therapy of PCa: to distinguish slow-growing tumors from aggressive tumors accurately and to identify resistance mechanisms to ongoing treatment. Radical prostatectomy or radiotherapy are the two main options for treatment with curative intent at the localized stage [2,3]. In disease progression, patients are usually treated by androgen deprivation (ADT) by either surgical castration or hormonal therapy via Luteinizing Hormone-Releasing Hormone (LHRH) analogs, as most of these tumors are hormone-sensitive (metastatic castration-sensitive prostate cancer, mCSPC) [3,4]. More recently, early treatment intensification by the addition of an androgen pathway inhibitor (ARPI) alone or combined with six cycles of docetaxel chemotherapy has been shown to improve outcomes [5]. Castration resistance occurs in virtually all of these patients after a median of 12 months following initial ADT, if given as monotherapy [6]. In the castration-resistant stage, median survival is about 3.5 years [3]. Depending on the therapy in the mCSPC setting, treatment options for metastatic castration-resistant prostate cancer (mCRPC) are based on performance status, symptoms, comorbidities, and molecular tumor characteristics. Therapeutic choices include abiraterone or enzalutamide, both inhibitors of the androgen receptor signaling pathway; chemotherapeutics (docetaxel or cabazitaxel); and prostate-specific membrane antigen (PSMA)-targeted radionuclide therapy with Lutetium-177 (Lu-177). Lu-177-PSMA radioligand therapy reduces disease progression and improves overall survival [7].

The mechanisms underlying the progression to castration resistance are not yet fully elucidated. In most patients, castration resistance is androgen receptor (*AR*)-driven, leading to reactivated *AR* signaling following ADT [8,9]. This is caused by increased *AR* expression via *AR* amplification, intratumoral androgen synthesis, *AR* mutation, or the expression of *AR* splice variants [10]. During prostate cancer development, tumor suppressor genes like *PTEN* or *TP53* are often inactivated by deletion and/or mutation, either as an early event or during disease progression, respectively [11]. The germline mutations in the genes *BRCA1*, *BRCA2*, *ATM*, or genes related to DNA mismatch repair are also associated with the early development of prostate cancer [1]. mCRPC patients with germline mutations in DNA repair genes (e.g., *BRCA1*, *BRCA2*) have been shown to benefit from the PARP-inhibitor olaparib [12,13]. In addition, copy number variations (CNV) and structural gene rearrangements are well-defined oncogenic drivers of PCa [1,14]. The mutational landscapes of mCSPC and mCRPC are similar. The common driver mutations in mCSPC occur in *TP53*, *SPOP* genes, and *WNT* (*CTNNB1*, *APC*, *RNF43*), cell cycle (*CDKN1B*, *CDKN2A*, *RB1*), and *PIk3K*/*AKT*/*mTOR* (*PTEN*, *PIK3R1*, *PIK3CA*) signaling pathways [15]. While *TP53* mutations in mCSPC are associated with more aggressive disease, *SPOP* mutations correlate with improved ADT plus *AR*-axis targeted therapy outcomes [15,16]. Since these genomic alterations provide new opportunities for the management and tailored treatment of patients with advanced PCa, there is a rapidly increasing demand for genomic testing by next-generation sequencing (NGS).

This study aimed to evaluate the utility and performance of a custom-designed NGS cfDNA panel based on the AmpliSeq HD Technology (ThermoFisher Scientific, Waltham, MA, USA) and to evaluate the diagnostic potential of this targeted approach in analyses of the liquid biopsies in mPCa. 

## 2. Materials and Methods

### 2.1. Patient Cohort

68 patients with advanced prostate cancer undergoing treatment at the University Hospital Basel or the St. Claraspital Basel in Switzerland were selected for this study. Detailed patient characteristics are summarized in Table 1 (see Section 3.2). Tissue specimens were collected and processed as part of the standard diagnostic routine. Briefly, tissue biopsies were collected in 10% buffered formalin and transferred to the Institute of Pathology. After embedding in paraffin, the formalin-fixed and paraffin-embedded (FFPE) blocks were cut and stained with Hematoxylin and Eosin (HE). A board-certified pathologist marked the tumor area designated for genomic analysis using NGS. This study was approved by the local ethical board (Ethikkommission Nordwestschweiz, EKNZ, Number 214-329). This study was performed in accordance with the Declaration of Helsinki. All patients gave written informed consent before participating in this study.

### 2.2. Liquid Biopsy Preparation

Blood samples were collected in EDTA-KE tubes (2 × 7.5 mL per patient). Plasma was separated from the blood cells by two consecutive rounds of centrifugation (1900× *g*, 10 min, 20 °C) within four hours after the blood withdrawal, transferred into four cryotubes, and stored at −80 °C until further processing. All blood samples were collected during the period between 2015 and 2022. In 35% (24/68) of patients, blood was collected within two months, in 20% (14/68) within one year, in 25% (17/68) more than a year from the initial biopsy, and in 20% (14/68) of patients, no data on the period between tissue biopsy and blood withdrawal were available.

### 2.3. cfDNA Extraction

Cell-free DNA (cfDNA) was extracted with the MagMAX Cell-Free Total Nucleic Acid Isolation Kit on the KingFisher DuoPrime instrument according to the manufacturer’s instructions (Thermo Fisher Scientific, Waltham, MA, USA). Briefly, 1.5 mL to 6 mL of plasma was incubated with Proteinase K and the Lysis/Binding solution, followed by the automated binding/washing, and elution steps. Elution was performed in 22 µL of the provided elution buffer, and the extracted cfDNA was stored at −20 °C.

### 2.4. DNA Extraction from Tissue Biopsies

Representative tumor areas on HE-stained tissue sections were scraped from the glass slide using a scalpel or razor blade and transferred to an Eppendorf tube. DNA extraction was performed using the Maxwell RSC FFPE Plus DNA Kit on the Maxwell RSC48 device (Promega, Madison, WI, USA) according to the manufacturer’s instructions. Briefly, proteinase K digestion was performed overnight at 70 °C. After adding the lysis buffer and mixing, the lysate was transferred to the Maxwell RSC Cartridge and placed into the corresponding deck tray of the Maxwell RSC instrument. The program was selected according to the Maxwell RSC FFPE Plus DNA Kit Technical Manual. DNA was eluted in 50 µL of nuclease-free water.

### 2.5. DNA Quantification

DNA and cfDNA were quantified with the Qubit dsDNA HS Assay Kit (Molecular Probes, Thermo Fisher Scientific, Waltham, MA, USA) on the Qubit Fluorometer 2.0 (Thermo Fisher Scientific, Waltham, MA, USA) according to the manufacturer’s instructions. 2 μL of DNA or cfDNA were incubated with 198 μL of working solution for 2 min at room temperature and then measured with the Qubit Fluorometer. Concentrations in ng/mL were calculated according to the manufacturer’s instructions. 

### 2.6. Custom-Designed AmpliSeq HD cfDNA Panel 

A targeted custom-designed cfDNA PCa Ion Torrent AmpliSeq HD gene panel for NGS was designed by selecting regions with recurrent mutations in prostate cancer-relevant genes from the public databases (cBioPortal, ClinVar, TCGA) and literature [11,17,18,19,20]. The panel comprised 273 amplicons of 46 genes with an amplicon size range of 75–140 bp (Appendix A). Hotspot regions in the following 46 genes were represented in this panel: *ACVR2A*, *AKT1*, *APC*, *AR*, *ASXL1*, *ATM*, *BRAF*, *BRCA1*, *BRCA2*, *CDK12*, *CHD1*, *CHEK2*, *CSMD3*, *CTNNB1*, *CUL3*, *CYP17A1*, *CYP19A1*, *FBXW7*, *FOXA1*, *HRAS*, *HSD17B4*, *HSD3B1*, *IDH1*, *JAK1*, *KDM6A*, *KMT2C*, *KRAS*, *LHCGR*, *MED12*, *MGA*, *MYC*, *NCOR1*, *NCOR2*, *PIK3CA*, *PIK3R1*, *PTEN*, *RB1*, *RNF43*, *SLCO1B1*, *SLCO1B3*, *SPOP*, *TP53*, *ZBTB16*, *ZFHX3*, *ZMYM3*, *ZNF780B*.

### 2.7. Spike-in Validation of Custom-Designed AmpliSeq HD Targeted cfDNA PCa Panel

To determine the specificity and sensitivity of the AmpliSeq HD custom-designed cfDNA PCa panel for the detection of somatic mutations, we designed four spiking “Allele Frequency” (AF) standards (AF 10%, AF 1%, AF 0.5%, and AF 0.1%) with the help of PCa cell lines. Genetic analysis of three human PCa cell lines (VCaP, LNCaP, and DU145) was performed with targeted sequencing with a commercial Oncomine Comprehensive v3 panel (Thermo Fisher Scientific, Waltham, MA, USA) to identify the mutations. LNCaP and DU145 cell lines were routinely passaged in Roswell Park Memorial Institute culture medium (RPMI 1640), and the VCaP cell line was maintained in Dulbecco’s Modified Eagle’s Medium (DMEM) medium. Culture media were supplemented with 10% (DU145, VCaP) or 20% (LNCaP) fetal bovine serum (Invitrogen, Waltham, MA, USA) and a 1% penicillin/streptomycin solution (BioConcept, Allschwil, Switzerland). Cell lines were cultured in a humidified incubator with 5% CO2 at 37 °C. All cell lines were cultured for at least two days after passaging, gently detached with Detachin (Genlantis, San Diego, CA, USA), and washed with phosphate-buffered saline (PBS, Gibco, Billings, MT, USA) to collect the cell pellet. DNA was isolated with a DNeasy kit (Qiagen, Hilden, Germany) according to the manufacturer’s instructions. Spiking standards were made by mixing DNA from all three cell lines into a wild-type DNA background (Promega, DNA, Madison, WI, USA) to the total DNA concentration of 15 ng/μL for each AF standard. Mutations identified in human PCa cell lines (VCaP, LNCaP, and DU145) were set to 10%, 1%, 0.5%, and 0.1% allele frequencies in retrospective AF standards. 22 ng of the total DNA amount was used for library preparation as described in the Section 2: Library preparation and sequencing.

### 2.8. Library Preparation and Sequencing

DNA originating from FFPE tissue was pre-treated with Uracil-DNA glycosylase (UDG, Thermo Fisher Scientific, Waltham, MA, USA). Depending on the DNA amount, targeted sequencing of the primary tumor was performed either with a custom-designed panel or with an alternative commercial panel (Oncomine BRCA1 and BRCA2 Panel and Oncomine Comprehensive v3 Panel from Thermo Fisher Scientific, Waltham, MA, USA) [21]. Library preparation of FFPE DNA was performed according to the manufacturer’s instructions (Thermo Fisher Scientific, Waltham, MA, USA). Libraries were purified with Agencourt AmpureXP beads (Beckman Coulter, Brea, CA, USA). The quantification of FFPE Libraries was performed with the Ion Universal Quantitation Kit (Thermo Fisher Scientific, Waltham, MA, USA). For cfDNA libraries, 25 ng was used as input library preparation for NGS, whenever possible. cfDNA libraries were purified with the Agencourt AmpureXP beads (Beckman Coulter, Brea, CA, USA) and quantified with the High Sensitivity D1000 Screen Tape kit on a TapeStation 4100 (Agilent, Santa Clara, CA, USA). FFPE libraries were diluted to 50 pM and cfDNA Libraries to 80 pM, loaded on Ion 550 chips by the Ion Chef instrument, and then sequenced on either an S5 Prime or an S5XL instrument (Thermo Fisher Scientific, Torrent Suite v5.16.1). To reach a theoretical LOD of 0.1% AF, a minimum of 20 ng of DNA and a minimum of 50,000× read depth per amplicon were used according to the manufacturer’s instructions.

### 2.9. NGS Data Analysis

Raw data were automatically processed on the Ion Torrent Server v5.16.1 and aligned to a hg19 reference genome using the Torrent Alignment Software (Torrent Suite v5.16.1). Only libraries that passed quality control cut-offs (FFPE: >95% on target reads, >90% uniformity, and >2000× average coverage; cfDNA: molecular uniformity > 90%, median reads per functional molecule > 7) were used for further analysis. The average base coverage depth for cfDNA libraries was 74,888 (median = 77,782), and the average molecular coverage was 2759.68 (min 1, max 95,788).

FFPE tumor and matched germline sequencing data were analyzed using PipeIT software [22]. cfDNA sequencing data were uploaded to the Ion Reporter Analysis Server v5.16 for variant calling. Molecular-tag-based sequencing based on AmpliSeq HD technology (Thermo Fisher Scientific, Waltham, MA, USA) has been used in this study. Variant calling in plasma cfDNA samples depends on the number of generated molecular families of reads, where each family has its own molecular tag. Therefore, the detection limit in this approach depends on the amount of material used in the library preparation and the read coverage depth. A valid tag family at a given DNA input was considered only if an average of 8 reads per amplicon for each input DNA molecule was obtained. A variant was considered a candidate somatic mutation only when all four of the following conditions were met: (a) the minimum number of reads required to call a variant is three, with at least one read from each DNA strand; (b) the variant were absent from public databases of common germline variants (1000 genomes, ExAC, gnomAD); (c) Variants with minor allele frequency (MAF) > 0.001 are considered likely benign and filtered out; (d) Variants that do not affect protein-coding regions (intronic, 3′, and 5′ untranslated region (UTR) variants) and synonymous mutations were filtered out; (e) predicted benign or likely benign variants (ClinVar) were filtered out. All detected variants have been manually inspected by the Integrative Genomics Viewer (v2.14.1).

### 2.10. Copy Number Analysis by Targeted cfDNA Panel

An estimation of panel-based *AR* copy number variation was performed at the gene level. The minimum gain threshold was determined via the CNV change distribution of AF spike standards positive for AR amplification, as determined by targeted sequencing previously. This analysis included two reference samples: Wild-Type (library prepared from commercial wild-type DNA used for the spiking experiment) and a male healthy donor (library prepared from a plasma cfDNA sample of the healthy donor). Amplicon coverage files obtained from the Torrent Server were used for CNV estimation. The mean and median of total reads for all AR amplicons vs. all other amplicons were calculated and compared. Ratios between means and medians were analyzed to determine the minimum threshold for the detection of AR amplification. The same approach was then applied to all patient plasma cfDNA samples to determine *AR* gains. Calculations of means and medians across all and *AR*-only amplicons were performed in R (3.5.1) in a Jupyter Notebook by reading CSV-formatted read count data tables from all patients and spike-in experiments.

### 2.11. Statistical Analysis

The Pearson correlation test tested the correlation between Prostate-Specific Antigen (PSA) level and cfDNA amount. The Wilcoxon–Mann–Whitney test was used to investigate the effect of ADT-ARSI therapy on cfDNA amounts in plasma. A *t*-test was used to test the impact of the hormonal status of PCa patients on the cfDNA amount. Fisher’s exact test has been performed to test dependencies between the number of ctDNA-positive patients and clinical parameters (treatment, hormonal status, tumor volume, and PSA level). *p*-values below 0.05 were considered statistically significant. Statistical analyses were performed using R packages v4.0.5. 

## 3. Results

### 3.1. Sensitivity of the Custom-Designed Targeted cfDNA PCa NGS Panel Based on AmpliSeq HD Technology

To assess the performance of our targeted sequencing approach, we generated four different spiked-in allelic frequency (AF) standards using PCa cell lines. Targeted sequencing of the three cell lines (DU145, LNCaP, and VCaP) revealed the presence of four mutations at 100% AF (Appendix A). The specificity of the panel was analyzed by mixing the DNA of three different cell lines (DU145, LNCaP, and VCaP) into the wild-type background. Spiked-in standards contained a mix of four mutations present in the AF standard at the following allelic frequencies: 10%, 1%, 0.5%, and 0.1%. All libraries were prepared with 22 ng of total DNA to reach the theoretical Limit of Detection (LOD) of 0.1% AF. While all four mutations were detected in the 10%, 1%, and 0.5% AF standards, only two were detected in the 0.1% AF standard (Figure 1). These data suggest that our approach’s sensitivity is 100% at an AF of 0.5%. However, mutations at lower AF can also be detected, but with significantly lower precision. 

### 3.2. Patient Cohort 

Plasma cfDNA sequencing was performed for 68 patients. A total of 63 patients had metastatic disease (cM1), while the remaining five had localized disease (cM0). Patient characteristics and tumor parameters are summarized in Table 1 and Appendix A. The mean age of the patients was 77 years. They were divided into two groups—high-volume disease and low-volume disease as defined in the CHAARTED study (high volume: ≥4 bone metastases, at least one outside the pelvis or spine, and/or visceral metastases; low volume: high volume criteria not met) [23]. According to this definition, 43 patients (63%) had a high-volume disease, and 24 (35%) had a low-volume disease. A total of 28 patients (41%) had mCSPC, and 40 patients (59%) had mCRPC.

### 3.3. Detection of Mutations in Tumor Biopsy and Matched cfDNA 

The mutation profile between the tumor (FFPE tissue) and liquid biopsy was compared for 42 mPCa patients. Driver mutations were detected by targeted sequencing in the tumor biopsies of 28 patients, and in 14 patients, no target mutation was identified. No target mutation was defined as either no mutation detected (n = 6/14) or a mutation detected in a tumor biopsy that is not covered by the custom AmpliSeq HD cfDNA Panel (n = 8/14). Library preparation failed only in three patient samples due to a low cfDNA amount (<2 ng), and these patients were excluded from further analyses. For detailed information, refer to Appendix A. In 21/28 (75%) patients, mutations from the primary tumor tissue were also detected in the plasma cfDNA (Table 2). Interestingly, mutations from primary tumors were detected in all tested plasma cfDNA from mCRPC (17/17) patients but only in 4/11 mCSPC patients (Table 2). A high discordance between plasma cfDNA and tumor biopsy DNA (7/11) was observed regarding mutation detection in the mCSPC group (Table 2). The majority of these patients (5/7) were under systemic therapy at the time of blood withdrawal, which might explain the high discordance. 

Two discordant mCSPC patients who were not under therapy at the time of blood withdrawal had both low volume disease and had either a low PSA amount (1.6 ng/mL, patient 46, Appendix A) or a high PSA amount (56.2 ng/mL, patient 141, Appendix A). Mutations in *AR*, *TP53*, *PTEN*, and *CHD1* genes were identified in 4/9 mCSPC patients (Appendix A) in plasma only but not in tumor tissue. After correction for the samples where no mutation was detected in the tumor tissue or detected mutations were not covered by the cfDNA PCa panel, the concordance rate between the matched tumor and liquid biopsy was 75%. AF of somatic variants strongly correlated with plasma cfDNA levels (Figure 2a). Remarkably, in most patients where the cfDNA amount was higher than 25 ng/mL, the AF of concordant somatic mutations between tumor and liquid biopsy was either as high as in tumor tissue or sometimes even higher (Figure 2b). 

### 3.4. Somatic Mutation Landscape in cfDNA in Advanced mCRPC and mCSPC PCa Patients

In addition to the matched FFPE-liquid biopsy cohort, 26 additional plasma samples (total n = 68) were included to analyze genetic alterations in cfDNA (Figure 3). LOD of 0.1% for variant calling in cfDNA samples was achieved in 81% (55/68) of patient samples. Sequencing with our custom-designed cfDNA AmpliSeq HD panel revealed somatic variants in 48 of 68 (71%) tested patients in recurrently mutated genes in PCa. The most recurrently mutated genes were *TP53* (62%), *SPOP* (19%), AR (19%), *CTNNB1* (12%), *PTEN* (12%), *RB1* (6%), and *FOXA1* (6%) (Figure 3). In 23/48 (48%) patients, we detected more than two somatic mutations in cfDNA (Figure 3). A detailed list of somatic variants detected in the matched cohort and cfDNA-only samples is presented in Appendix A. *BRCA1* and *BRCA2* mutations were also detected in our cohort. Patient 247 had *TP53*: c.743G > A (p.R248Q) (AF 5%); and *BRCA2*: c.8188G > A (p.A2730T) (AF 4%) mutations detected in both the FFPE biopsy and the cfDNA. The detection of somatic variants in cfDNA samples did not depend on the period between tumor and liquid biopsy, as most of the patients in the matched cohort had liquid biopsy sampling within one year after the tumor biopsy. 

### 3.5. Analytical Validation of the Copy Number Variation Detection

Accurate estimation of copy number variation is challenging in liquid biopsies. We tested the performance of our targeted panel-based approach in CNV estimation of *AR* gain on a reference set consisting of four different spike standards (AF 10%, AF 1%, AF 0.5%, and AF 0.1%) used as true positive and two reference samples used as true negative samples: WT (commercial wildtype DNA) and a healthy male donor (Figure 4a). Each standard mimics the sample with a 10%, 1%, 0.5%, and 0.1% ctDNA fraction. The analysis of median values for the total number of reads for all *AR* amplicons covered by the panel and the median value of the total number of reads for all other amplicons taken together demonstrated that *AR* amplification could be detected only in the AF 10% standard, where the median value of reads for *AR* amplicons (n = 119,842.8) was two times higher compared with the median value of total reads (n = 55,644) for all other amplicons. The assessment of genomic alterations in plasma cfDNA is heavily influenced by the fraction of ctDNA in plasma. AR amplification was not observed on AF 1%, 0.5%, and 0.1% standards or in negative reference samples (healthy or wt) due to the low ctDNA fraction. The ratio of 2 between two median values was detected in the *AR* 10% standard (Figure 4b) and therefore set as the threshold for analysis of *AR* amplification in patient samples. Analysis of *AR* amplifications in patients’ cfDNA samples showed *AR* gains in 13 patients, of which 12 were with mCRPC (Figure 4c). The same results were obtained if the mean values of total reads for AR and all other amplicons were used.

### 3.6. Correlation between PSA and cfDNA Amount in Plasma

PSA levels were not significantly higher in patients with the high-volume disease than in patients with the low-volume disease (mean/median: 480/58 ng/mL resp. 213/48 ng/mL, *t*-test *p* = 0.19) (Table 1). There was a positive correlation (R = 0.33, *p* = 0.0085) between cfDNA and PSA amount in plasma in our cohort (Figure 5a). The median PSA level in patients where the somatic mutation(s) were detected in cfDNA was 96.5 ng/mL, and in patients without detected mutation(s), it was 25.8 ng/mL. Therefore, these data indicate that the likelihood of detecting somatic mutation(s) in cfDNA were significantly associated with high PSA levels in mPCa patients (Figure 5b), which is in line with a previous study [24].

To further address this dependency, correlation analyses between PSA levels and ctDNA-positive samples have been performed (Table 3). ctDNA-positive samples were defined as samples where at least one somatic variant has been detected in plasma cfDNA. We observed a higher rate of ctDNA-positive plasma samples in patients with high PSA serum levels (>4 ng/mL) (Fisher’s Exact Test, *p* = 0.02).

### 3.7. Correlation between cfDNA Amount, the Onset of Therapy, and Hormonal Status

In 40/68 patients, ADT and/or androgen receptor signaling inhibitors (ARSI; abiraterone or enzalutamide), and in 4/68 patients, other therapy options (docetaxel, olaparib, LU177-PSMA, or carboplatin/paclitaxel) were administered shortly before or at the time of blood withdrawal (Appendix A). Of these 44 patients, 29 (66%) were at the stage of mCRPC, and 15 (34%) were at the stage of mCSPC. Wilcoxon–Mann–Whitney statistical testing was used to evaluate the difference in cfDNA amount between patients on therapy and patients not receiving any new treatment at the time of blood withdrawal. The amount of plasma cfDNA was significantly lower in the patients on the new treatment (Figure 6a). mCRPC patients did not have significantly higher cfDNA amounts in plasma than the mCSPC patients (Figure 6b, Two-Tailed *t*-test, *p* = 0.15). Likewise, there was no significant difference in ctDNA amounts in the plasma of mCRPC and mCSPC patients positive for the somatic variant (Figure 6c, Two-Tailed *t*-test, *p* = 0.097). mCRPC patients showed a trend toward a higher number of ctDNA-positive samples when compared to mCSPC, but significance was not reached (Table 3, Fisher’s Exact Test, *p* = 0.08). No correlation between the onset of therapy and tumor volume or the number of ctDNA-positive samples was observed (Table 3).

## 4. Discussion

In healthy individuals, plasma cfDNA mostly derives from white blood cells, whereas in cancer patients, a significant fraction of cfDNA originates from apoptotic and necrotic cells. Different physiological processes, such as inflammation, diabetes, tissue trauma, sepsis, and myocardial infarction, can influence cfDNA levels in the blood. The amount of cfDNA in the blood depends on tumor type and correlates directly to the stage of disease and tumor burden [25]. PCa is typically characterized by two-to-threefold higher cfDNA levels in plasma when compared to healthy subjects [9]. Moreover, levels of ctDNA can vary over time and be influenced by cancer progression and therapy. There is a growing demand for predictive or prognostic molecular testing in patients with mCRPC to individualize the sequence of treatment decisions and/or to direct patients to clinical trials [26,27,28]. In many of these patients, it is difficult, risky, or impossible to obtain new biopsies at the time of castration resistance, and tumor tissue from initial biopsies or resections is not always available or is insufficient for NGS analysis. In current clinical practice, cfDNA testing in patients with PCa requires centralized testing by commercial entities with pan-cancer cfDNA panels. However, solutions for in-house testing using cfDNA panels tailored to PCa are not yet in common use (Appendix A). To the best of our knowledge, besides a few reports employing the hybrid capture enrichment method for analysis of ctDNA by target, exome, or whole genome sequencing in PCa, this study is the first one to report on the applicability of amplicon-based sequencing in cfDNA analysis with the help of IonTorrent AmpliSeq HD technology in PCa patients [29,30]. A comprehensive evaluation of the analytical performance of several ctDNA assays using hybrid capture enrichment vs. amplicon sequencing assays (e.g., Thermo Fisher Scientific) indicated similar performance between these assays [31]. This study focused on the performance evaluation of our newly custom-designed AmpliSeq HD Panel for plasma cfDNA analysis in the mPCa cohort encompassing mCSPC and mCRPC patients. In accordance with previously published data, we observed a positive correlation between PSA and cfDNA amount in blood samples of mPCa patients [32]. Our study also provides evidence that the amount of cfDNA can serve as a surrogate for the ctDNA level. Our new PCa cfDNA panel detected somatic mutations in 75% of patients with matched tumors and a liquid biopsy. Others found a similar concordance rate between tissue biopsy and cfDNA in mCRPC patients by using a Foundation One Liquid Assay based on hybrid capture targeted sequencing of 70 genes (75.3%) or a liquid assay based on targeted sequencing of the exonic regions of 72 clinically relevant genes (75.6%) [9,33]. ctDNA levels in the blood are strong indicators for the response to first-line ARSI therapies and the overall survival of mCRPC [34]. There is evidence that cfDNA levels also differ between different lines of therapy in mCRPC, with a significant increase between first- and second-line and third- and fourth-line treatment of mCRPC [35]. We observed significantly lower cfDNA levels in the mPCa patients at the time of therapy (ADT and/or ARSI) compared to the mPCa patients who were not treated at the time of blood withdrawal. Similarly, ADT therapy decreases the ctDNA amount in mCSPC patients [36]. No significant difference in cfDNA level between mCRPC and mCSPC patients was observed in this study. 

In the setting of mCRPC, where molecular testing is currently most important, somatic mutations were detected in 88% (35/40) of all tested mCRPC patients. These data support the applicability of our custom-designed targeted ultra-deep cfDNA sequencing approach in routine clinical settings. The high discordance rate (7/11 cases) observed among mCSPC patients is likely due to lower cfDNA levels detected in the plasma of these patients due to systemic therapy initiated prior to blood withdrawal. This emphasizes the importance of the timing of blood sampling related to treatment initiation. In colorectal cancer, the onset of treatment with the multikinase inhibitor regorafenib causes the release of non-tumoral cfDNA into the blood, likely due to its toxic effect on normal tissue [37]. Therefore, blood for cfDNA testing should be drawn before a new treatment, e.g., ADT or ARSI, to not jeopardize the sensitivity of cfDNA mutation testing. One of the drawbacks of our study is the inability to estimate the ctDNA fraction in plasma samples, as we cannot precisely estimate copy number changes over the whole genome due to the limited panel size. However, our data provide evidence that the cfDNA plasma level represents a surrogate of the ctDNA fraction in patients with advanced PCa, which has been proposed as a strong prognostic marker in patients with PCa [34]. On the other hand, ultra-deep sequencing with a panel of smaller size is cost-effective, less error-prone (e.g., fewer false positive variant calls), and, therefore, easier to implement in the routine clinical setting [31]. There is a greater chance to detect ctDNA in mPCa patients with elevated PSA serum levels (>4 ng/mL) compared to mPCa patients with low PSA levels (<4 ng/mL) in the blood. This result is well in line with previous findings where a higher PSA level was strongly associated with a higher ctDNA fraction and, therefore, a higher likelihood of detecting targetable somatic variants [38]. To overcome the limitation of not being able to estimate the ctDNA fraction directly, one could use the combination of cfDNA and PSA amount as biomarkers for clinical decisions [35]. In cases of low cfDNA and PSA levels, genetic workup on a metastatic tumor biopsy should be attempted as a reflex test to avoid false negative results, if possible. Notably, rapid progression despite low PSA but high cfDNA amounts can occur in patients during ADT or ARSI, indicating transformation to AR-independent small cell carcinoma, which also needs to be confirmed by biopsy despite typical mutational patterns that might be found in cfDNA [39].

We identified the most frequent somatic mutations known to occur in mPCa (*AR*, *PTEN*, *SPOP*, *TP53*, *FOXA1*, and *BRCA2*). Remarkably, the variant allele fractions for shared mutations in tumor tissue and matched cfDNA were similar in most mPCa patients with high cfDNA amounts in plasma. A similar finding was previously reported for mCRPC patients [9]. As expected, mutations in the *TP53* gene were the most frequent ones identified among the patients in our cohort [40]. Interestingly, their AF ranged from 0.1% to 73%. In five patients (82FU, 115, 118, 141, and 188), *TP53* mutations with AF below 1% were detected in plasma cfDNA but not in the tumor tissue (Appendix A). Considering the low AF of detected *TP53* mutations (below 1% AF) and their absence from a tumor biopsy, these mutations might present CHIP (Clonal hematopoiesis of indeterminate potential)-associated somatic mutations [41]. To exclude reporting of CHIP-associated *TP53* mutations in the clinical routine, sequencing of buffy coat DNA should be considered in cases where detected AF is lower than 1%.

cfDNA analysis provides a non-invasive way to assess tumor somatic variants but has limited potential for detecting copy number variations. Importantly, our cfDNA biopsy assay demonstrated the possibility of detecting *AR* amplifications. *AR* amplification is an important mechanism of castration resistance and an adverse prognostic factor occurring in 40% of CRPC but almost never in CSPC [42]. We detected *AR* amplifications in 31% of mCRPC patients, in agreement with previous reports [43]. Due to the small size of our panel, gene losses could not be investigated, which is one of our assay’s limitations. However, reliable detection of gene losses of relevant genes (e.g., *PTEN*) remains challenging even when using larger panels [29,30]. Therefore, enforcing tissue analysis in the case of a negative cfDNA result should be considered, as recommended for other tumor types [44]. 

## 5. Conclusions

This study adds to the growing body of literature describing that cfDNA-targeted genomic analysis mirrors the somatic mutation landscape of PCa tumor biopsies and can be easily implemented in decision-making in routine clinical practice. We demonstrated that our targeted sequencing approach based on AmpliSeq HD technology provides a straightforward and cost-effective method for detecting clinically relevant somatic mutations in cfDNA, especially in mCRPC patients, to aid clinical decision-making. 

## Figures and Tables

**Figure 1 cancers-16-00045-f001:**
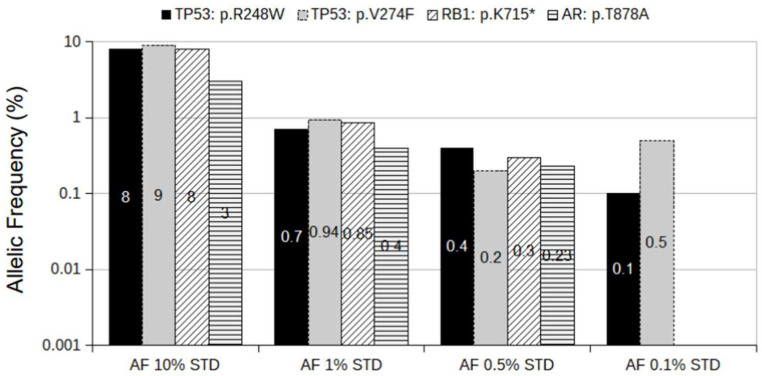
Spiked-in AF standards. The mixture of DNA from VCaP, LNCaP, and DU145 cell lines spiked into wild-type DNA background to AF of 10%, 1%, 0.5%, and 0.1% for the following mutations: *AR*: p.T878A; *RB1*: p.K715*; *TP53*: p.R248W; *TP53*: p.V274F. The *y*-axis is presented as a log(10) scale for better comparison between standards. The numbers in bars refer to the AF of the corresponding mutation.

**Figure 2 cancers-16-00045-f002:**
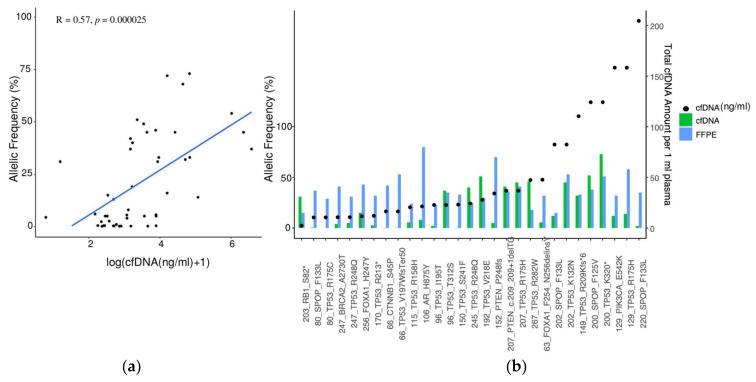
Concordance analysis of the matched tumor and liquid biopsy. (**a**) Correlation between AF and cfDNA amount in plasma (n = 48). Pearson correlation R = 0.57, *p* = 0.000025; (**b**) Concordance analysis of matched tumor biopsy and plasma cfDNA with regards to detected somatic variants. cfDNA amount (ng/mL) for each patient sample is indicated with a black dot. The *x*-axis displays the sample names, along with the corresponding gene names and observed amino-acid change.

**Figure 3 cancers-16-00045-f003:**
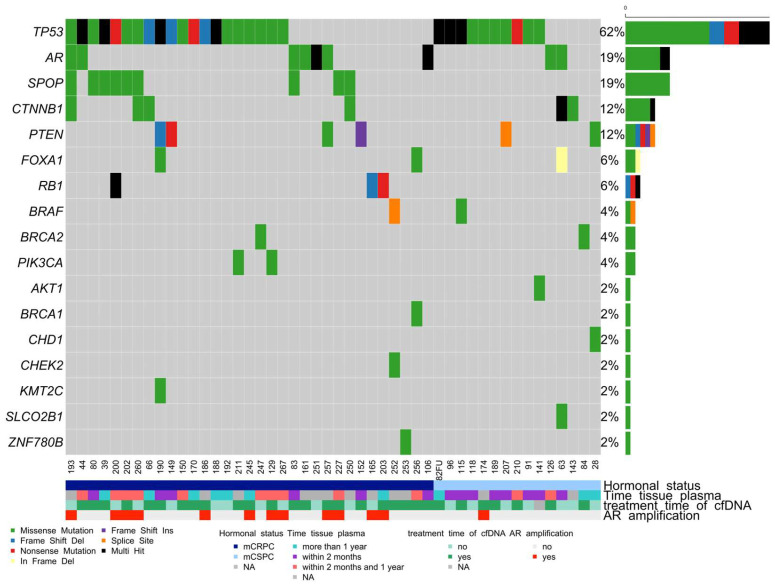
OncoPrint shows the distribution of genetic alterations in the cfDNA of mCRPC and mCSPC patients. Legend provides an overview of the types of genetic alterations in the genes (rows) of the individual samples (columns). Somatic mutations have been detected in 48 of the 68 cfDNA samples. Multi-hit refers to samples with more than one type of somatic variant.

**Figure 4 cancers-16-00045-f004:**
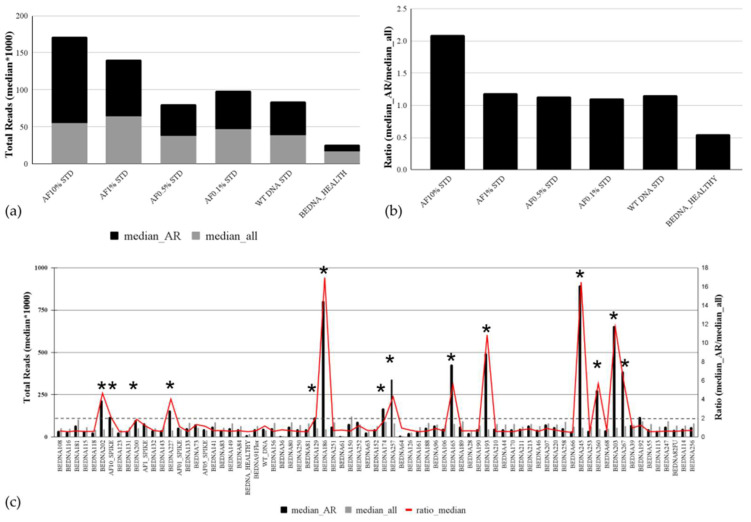
Copy number variation in the plasma cfDNA of mPCa patients. The analytical sensitivity of the targeted cfDNA panel for *AR* amplification detection is based on median values for total coverage for *AR* amplicons and all other amplicons of other genes covered by the panel (**a**) and on their ratio (**b**). (**c**) *AR* amplification analysis in cfDNA samples of 68 mPC patients. The median values of total reads for *AR* amplicons and the median value of the total number of reads for all other amplicons in all tested plasma cfDNA samples are presented in columns. The ratio of median values is shown with a red line. Samples with *AR* amplifications are indicated with an asterisk (*).

**Figure 5 cancers-16-00045-f005:**
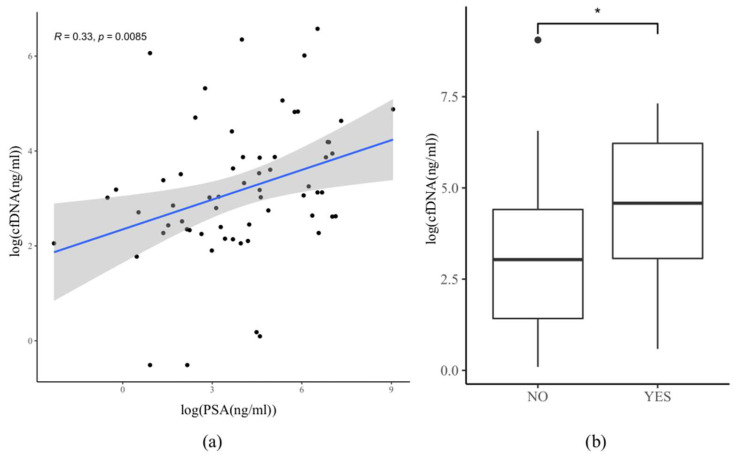
Correlation between PSA level and cfDNA amount in plasma. Data log(x) transformation were performed for better graphical visualization of the data due to the presence of outliers. (**a**) Correlation plot of pairwise estimates of cfDNA and PSA concentration in plasma (n = 62). *y*-axis: cfDNA concentration (ng/mL) in plasma. *x*-axis: PSA concentration (ng/mL) in serum. Pearson coefficient R = 0.33, *p* = 0.0085; (**b**) Correlation between PSA levels (ng/mL) and plasma cfDNA samples positive (YES) or negative (NO) for somatic mutation. Wilcoxon–Mann–Whitney test, *p* = 0.03 (*).

**Figure 6 cancers-16-00045-f006:**
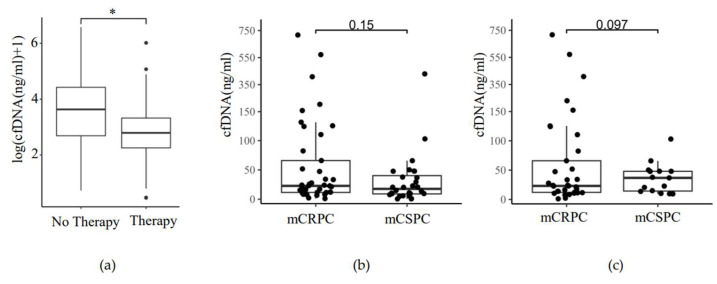
Plasma cfDNA level in response to ADT-ARSI therapy and hormonal status of mPCa. (**a**) ADT-ARSI therapy reduces plasma cfDNA in mPCa. Therapy: patients on ADT or ARSI therapy at the time or shortly before the blood withdrawal (n = 43). No therapy: patients without treatment at the time of blood withdrawal (n = 26). Wilcoxon–Mann–Whitney test, *p* = 0.031 (*); (**b**) cfDNA amount in plasma of mCRPC and mCSPC patients. *y*-axis: total cfDNA amount (ng/mL) in plasma. *x*-axis: Hormonal status (mCRPC (n = 40), mCSPC (n = 28)), Two-Tailed *t*-test, *p* = 0.15; (**c**) cfDNA amount in plasma of mCRPC and mCSPC patients positive for somatic variant. *y*-axis: total ctDNA amount (ng/mL) in plasma. *x*-axis: Hormonal status (mCRPC (n = 33), mCSPC (n = 15)), Two-Tailed *t*-test, *p* = 0.097.

**Table 1 cancers-16-00045-t001:** Clinicopathological characteristics of this study patients. The numbers refer to the number of patients (n = 68). ^1^ No data are available on disease volume in one patient; ^2^ No data are available in three patients, and in one patient, information on the volume is missing. ^3^ No PSA concentration data are available in 7 patients; ^4^ No data are available in one patient with regard to the disease volume; ^5^ Therapy at the time of cfDNA sampling: no data available on the treatment choice in one patient.

	Total	High Volume	Low Volume
**Number ^1^**	68	43	24
**Age mean value (*range*)**	77 (56–92)	75 (56–92)	78 (58–92)
**Gleason score (%) ^2^**	65 (100)	42	23
≤7 (%)	14 (19)	9	5
8 (%)	11 (16)	8	3
≥9 (%)	40 (59)	24	15
**PSA at the time of cfDNA (ng/mL) sampling: ^3^**	62	38	22
(median; 1st–3rd Quartile)	(57; 9.9–410)	(58; 14–335)	(48; 4–382)
**Disease status at the time of cfDNA ^4^**	68		
mCSPC (%)	28 (41)	14	14
mCRPC (%)	40 (59)	29	10
**Therapy at the time of cfDNA sampling (%) ^5^**	45 (66)	28	16

**Table 2 cancers-16-00045-t002:** Mutation concordance between tumor biopsy and plasma cfDNA in mPCa patients: “No mutation detected” is defined either as no mutation detected in the tumor tissue or as mutations detected in the tumor biopsy that the AmpliSeq HD cfDNA PCa Panel did not cover.

**Matched Samples** **FFPE—Liquid Biopsy**	**FFPE Tumor Biopsy** **No Mutation Detected**	**Concordant Cases** **FFPE—cfDNA**	**Discordant Cases** **FFPE—cfDNA**
mCRPC	5	17	0
mCSPC	9	4	7
Total	14	21	7

**Table 3 cancers-16-00045-t003:** Correlation between ctDNA-positive plasma samples and different clinical parameters. Numbers refer to the number of patients whose plasma tested positive for a somatic variant. The average AF is calculated across all somatic variants present in each defined group. All % values in the table have been rounded. Categorical variables are reported using counts of patients and were compared using the Fisher’s Exact test.

ClinicalParameter		Total Samples (n)	ctDNA Positive (n)	ctDNA Positive(%)	Average AF (%)	Fisher’s Exact Test *p* Value
**Treatment**	YES	43	29	67	12	0.14
	NO	22	19	86	17	
**Hormonal Status**	mCRPC	40	33	83	17	0.08
	mCSPC	25	15	60	10	
**Tumor Volume**	High	42	33	79	19	0.24
	Low	22	14	64	8	
**PSA**	Normal (<4 ng/mL)	8	3	38	14	0.02
	High	50	40	80	15	

## Data Availability

All data generated or analyzed during this study are included in this published article and the Appendix A.

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
