# Peer review of "Cell-Free DNA Genomic Profiling and Its Clinical Implementation in Advanced Prostate Cancer"

_cancers, 2023, doi:10.3390/cancers16010045_

Round 1

Reviewer 1 Report

Comments and Suggestions for Authors

The reviewed manuscript entitled: “Cell-Free DNA Genomic Profiling and its Clinical Implementation in Advanced Prostate Cancer” aimed to establish the utility and diagnostic potential of a bespoke panel for prostate cancer detection. The methodology applied in this study is appropriate and important controls are included. However, the presentation of the results can be improved, additional analyses are warranted, and caution should be taken in drawing conclusions on cfDNA plasma level as a surrogate to ctDNA fraction. CfDNA amounts can vary regardless of disease burden due to physiological and preanalytical condition, even if samples are processed in a standardized manner. In addition, the Authors mention in the discussion that ‘In colorectal cancer onset of treatment with the multikinase inhibitor regorafenib causes the release of non-tumoral cfDNA into the blood, likely due to its toxic effect on normal tissue [36].’ this example draws to an opposite conclusion as higher cfDNA levels would be expected on-treatment while this study reports on lower cfDNA levels. To be unbiased, discuss cfDNA plasma level as indication to ctDNA fraction in more detail.

Overall, the article is of interest to oncologists and beyond. Thus, I recommend publishing this manuscript subject to several additions/changes.

-        Abstract – the authors did not demonstrated / established ‘clinical utility’ (e.g., impacts of the test towards improved health outcomes). Suggest removing the word ‘clinical’ in sentence 40.

-        Methods –

1.      Suggest including information on the average molecular family’s coverage (including the average and range (rather than median)), similarly to the average base coverage depth in line 220-221.

2.      It’s a bit unclear if the cfDNA data was also filtered for germline variants in cases where tumors were unavailable? (e.g., for study IDs 91Flo and 186 – were the TP53 variants excluded from being germline?)

-        Results –

1.      Information is missing about how the theoretical LoD was evaluated (line 250). Similarly, Table S3 also missing information on how the ‘Achieved LOD’ is calculated.

2.      Figure 1 can be improved. Not sure why AF of 1% is used as a Y-axis baseline, and columns are presented downwards in STD 1%, 0.5%, and 0.1%. It is hard to evaluate which AFs (actual numbers) were recorded for each type of variant in the different STDs.

3.      Subheadings are missing their numbers after 3.2 (line 277 onwards).

4.      Suggest moving subsections ‘Detection of Mutations in Tumor Biopsy and matched cfDNA’ and ‘Somatic mutation landscape in cfDNA in advanced mCRPC and mCSPC PCa patients’ upwards, just after ‘3.2. Patient Cohort’ – for a more logical sequence of the results and highlighting ctDNA identification / findings, which is the focus of this paper.    

5.      Figure 2a – there is no explanation about the rational of using data transformation (log transformation, specifically log(x+1)).

6.      Correlations between ctDNA and clinical parameters/measurements should also be presented. From a brief look at the data presented in Table S3, it seems as the % of ctDNA-positive cases, and the average levels of ctDNA in ctDNA-positive cases (% allele frequency) are higher in (i) cases on therapy vs. no therapy, (ii) mCRPC vs. mCSPC, (iii) high- vs. low-volume, and maybe (iv) normal PSA levels (<4 ng/ml) vs. elevated. These additional analyses should be preforms and correlations to published data should be drawn.

Parameter

Total cases (n)

CtDNA positive (n)

ctDNA positive (%)

Average AF (%)

Treatment

On

43

28

65%

16%

No

22

17

77%

32%

Hormonal status

mCRPC

41

33

80%

24%

mCSPC

25

13

52%

16%

Tumor volume

High

43

31

72%

27%

Low

22

14

64%

11%

PSA

Normal

8 (7 of which mCSPC)

3

38%

14%

Elevated

50

37

74%

24%

7.      Figure 3b – I’m not convinced that there is a difference between the groups, the p-value is just under 0.05. Please provide a statistical justification for using one-tailed T-test instead of a two-tailed test. I believe that the p-value would be more significant when evaluating the ctDNA instead of cfDNA as mentioned above.

8.      Line 333 describes the discordant patient 46, but the discordant patient 141 is not explained.

9.      Figure 4b – to improve the visualization of the authors' point, consider arranging the x-axis according to increased amounts of cfDNA.

-        Discussion –

1.      As mentioned, I suggest that the Authors include a short paragraph in the Discussion section elaborating in an unbiased way about contributors to cfDNA plasma levels.

2.      Reference 30 (lines 413-415) describes a single amplicon sequencing panel (Thermo Fisher Scientific Oncomine cfDNA assay) and concludes similar performance to hybrid capture panels were identified. The part of the sentence ‘high sensitivity for detections of low-frequency mutations for amplicon sequencing panels’ should be removed.

-        Supplementary Table 3:

1.      Indicating a precise date of birth might considered an identifiable information and should be avoided. The Authors can instead include the age at biopsy, delta time between biopsy to blood draw etc.

2.      Instead of total reads, a raw coverage (before grouping into molecular families) is more informative as it factors the panel size (in bp).

Reviewer 2 Report

Comments and Suggestions for Authors

This study evaluated the utility and performance of a custom-designed NGS cfDNA panel based on the AmpliSeq HD Technology (ThermoFisher). This study provides a reference for clinical doctors to choose biomarkers detection methods for prostate cancer. The results of this study can accurately reflect the clinical performance of this panel, but some issues need to be addressed before it can be further considered.

1. Some abbreviations do not have corresponding English full-name explanations such as “PSA” (On page 5)

2. In section 3.1, the result can’t show the specificity of the panel. Please explain how the results demonstrate the specificity of this panel.

3. In section 3.1, only four mutations were detected to verify the sensitivity of this panel. I think more mutations should be detected to determine the sensitivity is 100% at an AF of 0.5%.

4. The number representation in the text includes both Arabic and English numerals, which need to be unified.

5. Some of the subheadings in the results section are not numbered.

Reviewer 3 Report

Comments and Suggestions for Authors

Investigating the genetic profiles in patients with metastatic castration-resistant or metastatic hormone-naïve prostate cancer could be very important for decision-making regarding treatment of advanced prostate cancer. The authors used AmpliSeq HD technology (amplicon-based sequencing) with a limited defined gene panel to identify somatic mutations. For this purpose, cell-free DNA was isolated from the blood of these patients. The aim of this rather methodical study was the usability of this sequencing method in clinical routine.

The analyzes could be of limited quality, especially if sufficient cfDNA could not be isolated from the blood sample obtained. As shown in Figure 2, the correlation between DNA amount and PSA value. A connection can be explained by active treatment, in which it was shown that the amount of cfDNA is different. The detection of mutations in the tumor biopsies and cfDNA from the same patient showed 100% agreement, but only if the patients were not on active systemic therapy and sufficient cfDNA could be isolated. Somatic mutation variants could be successfully detected using this method in the known mutated genes e.g. TP53, PTEN, AR, RB1, FOXA1. Analytical experiments to detect copy number variations revealed known limitations in the analyzes of liquid biopsies.

Genetic analysis are limited in decision making for therapy of the advanced prostate cancer and  such methods can be a key to the success of the treatment.
